# On the Convergence Rate of Decomposable Submodular Function Minimization

**Robert Nishihara,  Stefanie Jegelka,  Michael I. Jordan**
Electrical Engineering and Computer Science
University of California
Berkeley, CA 94720
{rkn,stefje,jordan}@eecs.berkeley.edu

## Abstract

Submodular functions describe a variety of discrete problems in machine learning, signal processing, and computer vision. However, minimizing submodular functions poses a number of algorithmic challenges. Recent work introduced an easy-to-use, parallelizable algorithm for minimizing submodular functions that decompose as the sum of "simple" submodular functions. Empirically, this algorithm performs extremely well, but no theoretical analysis was given. In this paper, we show that the algorithm converges linearly, and we provide upper and lower bounds on the rate of convergence. Our proof relies on the geometry of submodular polyhedra and draws on results from spectral graph theory.

## 1   Introduction

A large body of recent work demonstrates that many discrete problems in machine learning can be phrased as the optimization of a submodular set function [2]. A set function $F\colon 2^V \to \mathbb{R}$ over a ground set $V$ of $N$ elements is *submodular* if the inequality $F(A)+F(B) \geq F(A \cup B)+F(A \cap B)$ holds for all subsets $A, B \subseteq V$. Problems like clustering [33], structured sparse variable selection [1], MAP inference with higher-order potentials [28], and corpus extraction problems [31] can be reduced to the problem of submodular function minimization (SFM), that is

$$\min_{A \subseteq V} F(A). \tag{P1}$$

Although SFM is solvable in polynomial time, existing algorithms can be inefficient on large-scale problems. For this reason, the development of scalable, parallelizable algorithms has been an active area of research [24, 25, 29, 35]. Approaches to solving Problem (P1) are either based on combinatorial optimization or on convex optimization via the *Lovász extension*.

Functions that occur in practice are usually not arbitrary and frequently possess additional exploitable structure. For example, a number of submodular functions admit specialized algorithms that solve Problem (P1) very quickly. Examples include cut functions on certain kinds of graphs, concave functions of the cardinality $|A|$, and functions counting joint ancestors in trees. We will use the term *simple* to refer to functions $F$ for which we have a fast subroutine for minimizing $F + s$, where $s \in \mathbb{R}^N$ is any modular function. We treat these subroutines as black boxes. Many commonly occuring submodular functions (for example, graph cuts, hypergraph cuts, MAP inference with higher-order potentials [16, 28, 37], co-segmentation [22], certain structured-sparsity inducing functions [26], covering functions [35], and combinations thereof) can be expressed as a sum

$$F(A) = \sum\nolimits_{r=1}^{R} F_r(A) \tag{1}$$

of simple submodular functions. Recent work demonstrates that this structure offers important practical benefits [25, 29, 35]. For instance, it admits iterative algorithms that minimize each $F_r$ separately and combine the results in a straightforward manner (for example, dual decomposition).

In particular, it has been shown that the minimization of decomposable functions can be rephrased as a *best-approximation problem*, the problem of finding the closest points in two convex sets [25]. This formulation brings together SFM and classical projection methods and yields empirically fast, parallel, and easy-to-implement algorithms. In these cases, the performance of projection methods depends heavily on the specific geometry of the problem at hand and is not well understood in general. Indeed, while Jegelka et al. [25] show good empirical results, the analysis of this alternative approach to SFM was left as an open problem.

**Contributions.** In this work, we study the geometry of the submodular best-approximation problem and ground the prior empirical results in theoretical guarantees. We show that SFM via alternating projections, or block coordinate descent, converges at a *linear rate*. We show that this rate holds for the best-approximation problem, relaxations of SFM, and the original discrete problem. More importantly, we prove upper and lower bounds on the worst-case rate of convergence. Our proof relies on analyzing angles between the polyhedra associated with submodular functions and draws on results from spectral graph theory. It offers insight into the geometry of submodular polyhedra that may be beneficial beyond the analysis of projection algorithms.

**Submodular minimization.** The first polynomial-time algorithm for minimizing arbitrary submodular functions was a consequence of the ellipsoid method [19]. Strongly and weakly polynomial-time combinatorial algorithms followed [32]. The current fastest running times are $O(N^5\tau_1 + N^6)$ [34] in general and $O((N^4\tau_1 + N^5)\log F_{\max})$ for integer-valued functions [23], where $F_{\max} = \max_A |F(A)|$ and $\tau_1$ is the time required to evaluate $F$. Some work has addressed decomposable functions [25, 29, 35]. The running times in [29] apply to integer-valued functions and range from $O((N + R)^2 \log F_{\max})$ for cuts to $O((N + Q^2 R)(N + Q^2 R + QR\tau_2)\log F_{\max})$, where $Q \le N$ is the maximal cardinality of the support of any $F_r$, and $\tau_2$ is the time required to minimize a simple function. Stobbe and Krause [35] use a convex optimization approach based on Nesterov's smoothing technique. They achieve a (sublinear) convergence rate of $O(1/k)$ for the discrete SFM problem. Their results and our results do not rely on the function being integral.

**Projection methods.** Algorithms based on alternating projections between convex sets (and related methods such as the Douglas–Rachford algorithm) have been studied extensively for solving convex feasibility and best-approximation problems [4, 5, 7, 11, 12, 20, 21, 36, 38]. See Deutsch [10] for a survey of applications. In the simple case of subspaces, the convergence of alternating projections has been characterized in terms of the Friedrichs angle $c_F$ between the subspaces [5, 6]. There are often good ways to compute $c_F$ (see Lemma 6), which allow us to obtain concrete linear rates of convergence for subspaces. The general case of alternating projections between arbitrary convex sets is less well understood. Bauschke and Borwein [3] give a general condition for the linear convergence of alternating projections in terms of the value $\kappa_*$ (defined in Section 3.1). However, except in very limited cases, it is unclear how to compute or even bound $\kappa_*$. While it is known that $\kappa_* < \infty$ for polyhedra [5, Corollary 5.26], the rate may be arbitrarily slow, and the challenge is to bound the linear rate away from one. We are able to give a specific *uniform* linear rate for the submodular polyhedra that arise in SFM.

Although both $\kappa_*$ and $c_F$ are useful quantities for understanding the convergence of projection methods, they largely have been studied independently of one another. In this work, we relate these two quantities for polyhedra, thereby obtaining some of the generality of $\kappa_*$ along with the computability of $c_F$. To our knowledge, we are the first to relate $\kappa_*$ and $c_F$ outside the case of subspaces. We feel that this connection may be useful beyond the context of submodular polyhedra.

## 1.1  Background

Throughout this paper, we assume that $F$ is a sum of simple submodular functions $F_1, \ldots, F_R$ and that $F(\emptyset) = 0$. Points $s \in \mathbb{R}^N$ can be identified with (modular) set functions via $s(A) = \sum_{n \in A} s_n$. The *base polytope* of $F$ is defined as the set of all modular functions that are dominated by $F$ and that sum to $F(V)$,

$$B(F) = \{s \in \mathbb{R}^N \mid s(A) \le F(A) \text{ for all } A \subseteq V \text{ and } s(V) = F(V)\}.$$

The *Lovász extension* $f \colon \mathbb{R}^N \to \mathbb{R}$ of $F$ can be written as the support function of the base polytope, that is $f(x) = \max_{s \in B(F)} s^\top x$. Even though $B(F)$ may have exponentially many faces, the extension $f$ can be evaluated in $O(N \log N)$ time [15]. The discrete SFM problem (P1) can be relaxed to

the non-smooth convex optimization problem

$$\min_{x \in [0,1]^N} f(x) \quad \equiv \quad \min_{x \in [0,1]^N} \sum_{r=1}^{R} f_r(x), \tag{P2}$$

where $f_r$ is the Lovász extension of $F_r$. This relaxation is exact – rounding an optimal continuous solution yields the indicator vector of an optimal discrete solution. The formulation in Problem (P2) is amenable to dual decomposition [30] and smoothing techniques [35], but suffers from the non-smoothness of $f$ [25]. Alternatively, we can formulate a proximal version of the problem

$$\min_{x \in \mathbb{R}^N} f(x) + \tfrac{1}{2}\|x\|^2 \quad \equiv \quad \min_{x \in \mathbb{R}^N} \sum_{r=1}^{R} (f_r(x) + \tfrac{1}{2R}\|x\|^2). \tag{P3}$$

By thresholding the optimal solution of Problem (P3) at zero, we recover the indicator vector of an optimal discrete solution [17], [2, Proposition 8.4].

**Lemma 1.** *[25] The dual of the right-hand side of Problem* (P3) *is the best-approximation problem*

$$\min \|a - b\|^2 \quad a \in \mathcal{A}, \ b \in \mathcal{B}, \tag{P4}$$

*where* $\mathcal{A} = \{(a_1, \ldots, a_R) \in \mathbb{R}^{NR} \mid \sum_{r=1}^{R} a_r = 0\}$ *and* $\mathcal{B} = B(F_1) \times \cdots \times B(F_R)$.

Lemma 1 implies that we can minimize a decomposable submodular function by solving Problem (P4), which means finding the closest points between the subspace $\mathcal{A}$ and the product $\mathcal{B}$ of base polytopes. Projecting onto $\mathcal{A}$ is straightforward because $\mathcal{A}$ is a subspace. Projecting onto $\mathcal{B}$ amounts to projecting onto each $B(F_r)$ separately. The projection $\Pi_{B(F_r)} z$ of a point $z$ onto $B(F_r)$ may be solved by minimizing $F_r - z$ [25]. We can compute these projections easily because each $F_r$ is simple.

Throughout this paper, we use $\mathcal{A}$ and $\mathcal{B}$ to refer to the specific polyhedra defined in Lemma 1 (which live in $\mathbb{R}^{NR}$) and $P$ and $Q$ to refer to general polyhedra (sometimes arbitrary convex sets) in $\mathbb{R}^D$. Note that the polyhedron $\mathcal{B}$ depends on the submodular functions $F_1, \ldots, F_R$, but we omit the dependence to simplify our notation. Our bound will be uniform over all submodular functions.

## 2   Algorithm and Idea of Analysis

A popular class of algorithms for solving best-approximation problems are projection methods [5]. The most straightforward approach uses alternating projections (AP) or block coordinate descent. Start with any point $a_0 \in \mathcal{A}$, and inductively generate two sequences via $b_k = \Pi_{\mathcal{B}} a_k$ and $a_{k+1} = \Pi_{\mathcal{A}} b_k$. Given the nature of $\mathcal{A}$ and $\mathcal{B}$, this algorithm is easy to implement and use in our setting, and it solves Problem (P4) [25]. This is the algorithm that we will analyze.

The sequence $(a_k, b_k)$ will eventually converge to an optimal pair $(a_*, b_*)$. We say that AP converges linearly with rate $\alpha < 1$ if $\|a_k - a_*\| \le C_1 \alpha^k$ and $\|b_k - b_*\| \le C_2 \alpha^k$ for all $k$ and for some constants $C_1$ and $C_2$. Smaller values of $\alpha$ are better.

**Analysis: Intuition.** We will provide a detailed analysis of the convergence of AP for the polyhedra $\mathcal{A}$ and $\mathcal{B}$. To motivate our approach, we first provide some intuition with the following much-simplified setup. Let $U$ and $V$ be one-dimensional subspaces spanned by the unit vectors $u$ and $v$ respectively. In this case, it is known that AP converges linearly with rate $\cos^2 \theta$, where $\theta \in [0, \frac{\pi}{2}]$ is the angle such that $\cos \theta = u^\top v$. The smaller the angle, the slower the rate of convergence. For subspaces $U$ and $V$ of higher dimension, the relevant generalization of the "angle" between the subspaces is the *Friedrichs angle* [11, Definition 9.4], whose cosine is given by

$$c_F(U, V) = \sup \left\{ u^\top v \mid u \in U \cap (U \cap V)^\perp, v \in V \cap (U \cap V)^\perp, \|u\| \le 1, \|v\| \le 1 \right\}. \tag{2}$$

In finite dimensions, $c_F(U, V) < 1$. In general, when $U$ and $V$ are subspaces of arbitrary dimension, AP will converge linearly with rate $c_F(U, V)^2$ [11, Theorem 9.8]. If $U$ and $V$ are *affine spaces*, AP still converges linearly with rate $c_F(U - u, V - v)^2$, where $u \in U$ and $v \in V$.

We are interested in rates for *polyhedra* $P$ and $Q$, which we define as the intersection of finitely many halfspaces. We generalize the preceding results by considering all pairs $(P_x, Q_y)$ of

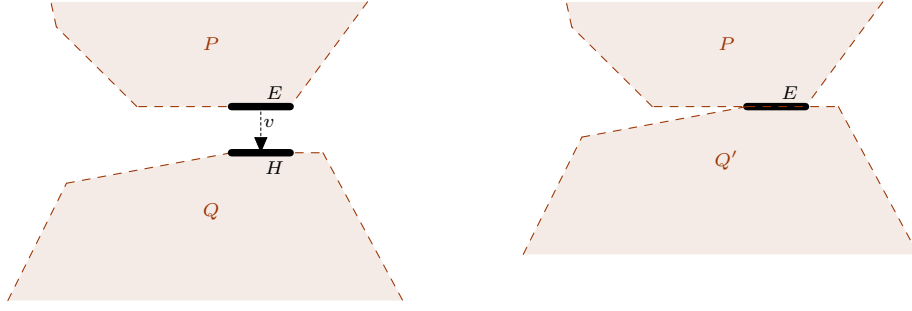

Figure 1: The optimal sets $E$, $H$ in Equation (4), the vector $v$, and the shifted polyhedron $Q'$.

faces of $P$ and $Q$ and showing that the convergence rate of AP between $P$ and $Q$ is at worst $\max_{x,y} c_F(\mathrm{aff}_0(P_x), \mathrm{aff}_0(Q_y))^2$, where $\mathrm{aff}(C)$ is the affine hull of $C$ and $\mathrm{aff}_0(C) = \mathrm{aff}(C) - c$ for some $c \in C$. The faces $\{P_x\}_{x \in \mathbb{R}^D}$ of $P$ are defined as the nonempty maximizers of linear functions over $P$, that is

$$P_x = \arg\max_{p \in P} x^\top p. \tag{3}$$

While we look at angles between pairs of faces, we remark that Deutsch and Hundal [13] consider a different generalization of the "angle" between arbitrary convex sets.

**Roadmap of the Analysis.** Our analysis has two main parts. First, we relate the convergence rate of AP between polyhedra $P$ and $Q$ to the angles between the faces of $P$ and $Q$. To do so, we give a general condition under which AP converges linearly (Theorem 2), which we show depends on the angles between the faces of $P$ and $Q$ (Corollary 5) in the polyhedral case. Second, we specialize to the polyhedra $\mathcal{A}$ and $\mathcal{B}$, and we equate the angles with eigenvalues of certain matrices and use tools from spectral graph theory to bound the relevant eigenvalues in terms of the conductance of a specific graph. This yields a worst-case bound of $1 - \frac{1}{N^2 R^2}$ on the rate, stated in Theorem 12.

In Theorem 14, we show a lower bound of $1 - \frac{2\pi^2}{N^2 R}$ on the worst-case convergence rate.

## 3 The Upper Bound

We first derive an upper bound on the rate of convergence of AP between the polyhedra $\mathcal{A}$ and $\mathcal{B}$. The results in this section are proved in Appendix A.

### 3.1 A Condition for Linear Convergence

We begin with a condition under which AP between two closed convex sets $P$ and $Q$ converges linearly. This result is similar to that of Bauschke and Borwein [3, Corollary 3.14], but the rate we achieve is twice as fast and relies on slightly weaker assumptions.

We will need a few definitions from Bauschke and Borwein [3]. Let $d(K_1, K_2) = \inf\{\|k_1 - k_2\| : k_1 \in K_1, k_2 \in K_2\}$ be the distance between sets $K_1$ and $K_2$. Define the sets of "closest points" as

$$E = \{p \in P \mid d(p, Q) = d(P, Q)\} \qquad H = \{q \in Q \mid d(q, P) = d(Q, P)\}, \tag{4}$$

and let $v = \Pi_{Q-P} 0$ (see Figure 1). Note that $H = E + v$, and when $P \cap Q \neq \emptyset$ we have $v = 0$ and $E = H = P \cap Q$. Therefore, we can think of the pair $(E, H)$ as a generalization of the intersection $P \cap Q$ to the setting where $P$ and $Q$ do not intersect. Pairs of points $(e, e + v) \in E \times H$ are solutions to the best-approximation problem between $P$ and $Q$. In our analysis, we will mostly study the translated version $Q' = Q - v$ of $Q$ that intersects $P$ at $E$.

For $x \in \mathbb{R}^D \backslash E$, the function $\kappa$ relates the distance to $E$ with the distances to $P$ and $Q'$,

$$\kappa(x) = \frac{d(x, E)}{\max\{d(x, P), d(x, Q')\}}.$$

If $\kappa$ is bounded, then whenever $x$ is close to both $P$ and $Q'$, it must also be close to their intersection. If, for example, $D \geq 2$ and $P$ and $Q$ are balls of radius one whose centers are separated by distance

exactly two, then $\kappa$ is unbounded. The maximum $\kappa_* = \sup_{x \in (P \cup Q') \backslash E} \kappa(x)$ is useful for bounding the convergence rate.

**Theorem 2.** *Let $P$ and $Q$ be convex sets, and suppose that $\kappa_* < \infty$. Then AP between $P$ and $Q$ converges linearly with rate $1 - \frac{1}{\kappa_*^2}$. Specifically,*

$$\|p_k - p_*\| \le 2\|p_0 - p_*\|(1 - \tfrac{1}{\kappa_*^2})^k \quad and \quad \|q_k - q_*\| \le 2\|q_0 - q_*\|(1 - \tfrac{1}{\kappa_*^2})^k.$$

### 3.2 Relating $\kappa_*$ to the Angles Between Faces of the Polyhedra

In this section, we consider the case of polyhedra $P$ and $Q$, and we bound $\kappa_*$ in terms of the angles between pairs of their faces. In Lemma 3, we show that $\kappa$ is nondecreasing along the sequence of points generated by AP between $P$ and $Q'$. We treat points $p$ for which $\kappa(p) = 1$ separately because those are the points from which AP between $P$ and $Q'$ converges in one step. This lemma enables us to bound $\kappa(p)$ by initializing AP at $p$ and bounding $\kappa$ at some later point in the resulting sequence.

**Lemma 3.** *For any $p \in P \backslash E$, either $\kappa(p) = 1$ or $1 < \kappa(p) \le \kappa(\Pi_{Q'} p)$. Similarly, for any $q \in Q' \backslash E$, either $\kappa(q) = 1$ or $1 < \kappa(q) \le \kappa(\Pi_P q)$.*

We can now bound $\kappa$ by angles between faces of $P$ and $Q$.

**Proposition 4.** *If $P$ and $Q$ are polyhedra and $p \in P \backslash E$, then there exist faces $P_x$ and $Q_y$ such that*

$$1 - \frac{1}{\kappa(p)^2} \le c_F(\mathrm{aff}_0(P_x), \mathrm{aff}_0(Q_y))^2.$$

*The analogous statement holds when we replace $p \in P \backslash E$ with $q \in Q' \backslash E$.*

Note that $\mathrm{aff}_0(Q_y) = \mathrm{aff}_0(Q'_y)$. Proposition 4 immediately gives us the following corollary.

**Corollary 5.** *If $P$ and $Q$ are polyhedra, then*

$$1 - \frac{1}{\kappa_*^2} \le \max_{x,y \in \mathbb{R}^D} c_F(\mathrm{aff}_0(P_x), \mathrm{aff}_0(Q_y))^2.$$

### 3.3 Angles Between Subspaces and Singular Values

Corollary 5 leaves us with the task of bounding the Friedrichs angle. To do so, we first relate the Friedrichs angle to the singular values of certain matrices in Lemma 6. We then specialize this to base polyhedra of submodular functions. For convenience, we prove Lemma 6 in Appendix A.5, though this result is implicit in the characterization of principal angles between subspaces given in [27, Section 1]. Ideas connecting angles between subspaces and eigenvalues are also used by Diaconis et al. [14].

**Lemma 6.** *Let $S$ and $T$ be matrices with orthonormal rows and with equal numbers of columns. If all of the singular values of $ST^\top$ equal one, then $c_F(\mathrm{null}(S), \mathrm{null}(T)) = 0$. Otherwise, $c_F(\mathrm{null}(S), \mathrm{null}(T))$ is equal to the largest singular value of $ST^\top$ that is less than one.*

**Faces of relevant polyhedra.** Let $\mathcal{A}_x$ and $\mathcal{B}_y$ be faces of the polyhedra $\mathcal{A}$ and $\mathcal{B}$ from Lemma 1. Since $\mathcal{A}$ is a vector space, its only nonempty face is $\mathcal{A}_x = \mathcal{A}$. Hence, $\mathcal{A}_x = \mathrm{null}(S)$, where $S$ is an $N \times NR$ matrix of $N \times N$ identity matrices $I_N$:

$$S = \frac{1}{\sqrt{R}} \left( \underbrace{\begin{matrix} I_N & \cdots & I_N \end{matrix}}_{\text{repeated } R \text{ times}} \right). \tag{5}$$

The matrix for $\mathrm{aff}_0(\mathcal{B}_y)$ requires a bit more elaboration. Since $\mathcal{B}$ is a Cartesian product, we have $\mathcal{B}_y = B(F_1)_{y_1} \times \cdots \times B(F_R)_{y_R}$, where $y = (y_1, \ldots, y_R)$ and $B(F_r)_{y_r}$ is a face of $B(F_r)$. To proceed, we use the following characterization of faces of base polytopes [2, Proposition 4.7].

**Proposition 7.** *Let $F$ be a submodular function, and let $B(F)_x$ be a face of $B(F)$. Then there exists a partition of $V$ into disjoint sets $A_1, \ldots, A_M$ such that*

$$\mathrm{aff}(B(F)_x) = \bigcap_{m=1}^M \{s \in \mathbb{R}^N \mid s(A_1 \cup \cdots \cup A_m) = F(A_1 \cup \cdots \cup A_m)\}.$$

The following corollary is immediate.

**Corollary 8.** *Define $F$, $B(F)_x$, and $A_1, \ldots, A_M$ as in Proposition 7. Then*

$$\text{aff}_0(B(F)_x) = \bigcap_{m=1}^{M} \{s \in \mathbb{R}^N \mid s(A_1 \cup \cdots \cup A_m) = 0\}.$$

By Corollary 8, for each $F_r$, there exists a partition of $V$ into disjoint sets $A_{r1}, \ldots, A_{rM_r}$ such that

$$\text{aff}_0(\mathcal{B}_y) = \bigcap_{r=1}^{R} \bigcap_{m=1}^{M_r} \{(s_1, \ldots, s_R) \in \mathbb{R}^{NR} \mid s_r(A_{r1} \cup \cdots \cup A_{rm}) = 0\}. \tag{6}$$

In other words, we can write $\text{aff}_0(\mathcal{B}_y)$ as the nullspace of either of the matrices

$$T' = \begin{pmatrix} 1_{A_{11}}^\top & & & \\ \vdots & & & \\ 1_{A_{11} \cup \cdots \cup A_{1M_1}}^\top & & & \\ & \ddots & & \\ & & 1_{A_{R1}}^\top & \\ & & \vdots & \\ & & 1_{A_{R1} \cup \cdots \cup A_{RM_R}}^\top \end{pmatrix} \quad \text{or} \quad T = \begin{pmatrix} \frac{1_{A_{11}}^\top}{\sqrt{|A_{11}|}} & & & \\ \vdots & & & \\ \frac{1_{A_{1M_1}}^\top}{\sqrt{|A_{1M_1}|}} & & & \\ & \ddots & & \\ & & \frac{1_{A_{R1}}^\top}{\sqrt{|A_{R1}|}} & \\ & & \vdots & \\ & & \frac{1_{A_{RM_R}}^\top}{\sqrt{|A_{RM_R}|}} \end{pmatrix},$$

where $1_A$ is the indicator vector of $A \subseteq V$. For $T'$, this follows directly from Equation (6). $T$ can be obtained from $T'$ via left multiplication by an invertible matrix, so $T$ and $T'$ have the same nullspace. Lemma 6 then implies that $c_F(\text{aff}_0(\mathcal{A}_x), \text{aff}_0(\mathcal{B}_y))$ equals the largest singular value of

$$ST^\top = \frac{1}{\sqrt{R}} \begin{pmatrix} \frac{1_{A_{11}}}{\sqrt{|A_{11}|}} & \cdots & \frac{1_{A_{1M_1}}}{\sqrt{|A_{1M_1}|}} & \cdots & \frac{1_{A_{R1}}}{\sqrt{|A_{R1}|}} & \cdots & \frac{1_{A_{RM_R}}}{\sqrt{|A_{RM_R}|}} \end{pmatrix}$$

that is less than one. We rephrase this conclusion in the following remark.

**Remark 9.** *The largest eigenvalue of $(ST^\top)^\top(ST^\top)$ less than one equals $c_F(\text{aff}_0(\mathcal{A}_x), \text{aff}_0(\mathcal{B}_y))^2$.*

Let $M_{\text{all}} = M_1 + \cdots + M_R$. Then $(ST^\top)^\top(ST^\top)$ is the $M_{\text{all}} \times M_{\text{all}}$ square matrix whose rows and columns are indexed by $(r, m)$ with $1 \le r \le R$ and $1 \le m \le M_r$ and whose entry corresponding to row $(r_1, m_1)$ and column $(r_2, m_2)$ equals

$$\frac{1}{R} \frac{1_{A_{r_1 m_1}}^\top 1_{A_{r_2 m_2}}}{\sqrt{|A_{r_1 m_1}||A_{r_2 m_2}|}} = \frac{1}{R} \frac{|A_{r_1 m_1} \cap A_{r_2 m_2}|}{\sqrt{|A_{r_1 m_1}||A_{r_2 m_2}|}}.$$

### 3.4 Bounding the Relevant Eigenvalues

It remains to bound the largest eigenvalue of $(ST^\top)^\top(ST^\top)$ that is less than one. To do so, we view the matrix in terms of the symmetric normalized Laplacian of a weighted graph. Let $G$ be the graph whose vertices are indexed by $(r, m)$ with $1 \le r \le R$ and $1 \le m \le M_r$. Let the edge between vertices $(r_1, m_1)$ and $(r_2, m_2)$ have weight $|A_{r_1 m_1} \cap A_{r_2 m_2}|$. We may assume that $G$ is connected (the analysis in this case subsumes the analysis in the general case). The symmetric normalized Laplacian $\mathcal{L}$ of this graph is closely related to our matrix of interest,

$$(ST^\top)^\top(ST^\top) = I - \tfrac{R-1}{R}\mathcal{L}. \tag{7}$$

Hence, the largest eigenvalue of $(ST^\top)^\top(ST^\top)$ that is less than one can be determined from the smallest nonzero eigenvalue $\lambda_2(\mathcal{L})$ of $\mathcal{L}$. We bound $\lambda_2(\mathcal{L})$ via Cheeger's inequality (stated in Appendix A.6) by bounding the Cheeger constant $h_G$ of $G$.

**Lemma 10.** *For $R \ge 2$, we have $h_G \ge \frac{2}{NR}$ and hence $\lambda_2(\mathcal{L}) \ge \frac{2}{N^2 R^2}$.*

We prove Lemma 10 in Appendix A.7. Combining Remark 9, Equation (7), and Lemma 10, we obtain the following bound on the Friedrichs angle.

**Proposition 11.** *Assuming that $R \geq 2$, we have*

$$c_F(\text{aff}_0(\mathcal{A}_x), \text{aff}_0(\mathcal{B}_y))^2 \leq 1 - \frac{R-1}{R}\frac{2}{N^2R^2} \leq 1 - \frac{1}{N^2R^2}.$$

Together with Theorem 2 and Corollary 5, Proposition 11 implies the final bound on the rate.

**Theorem 12.** *The AP algorithm for Problem* (P4) *converges linearly with rate* $1 - \frac{1}{N^2R^2}$, *i.e.,*

$$\|a_k - a_*\| \leq 2\|a_0 - a_*\|(1 - \tfrac{1}{N^2R^2})^k \quad and \quad \|b_k - b_*\| \leq 2\|b_0 - b_*\|(1 - \tfrac{1}{N^2R^2})^k.$$

## 4 A Lower Bound

To probe the tightness of Theorem 12, we construct a "bad" submodular function and decomposition that lead to a slow rate. Appendix B gives the formal details. Our example is an augmented cut function on a cycle: for each $x, y \in V$, define $G_{xy}$ to be the cut function of a single edge $(x, y)$,

$$G_{xy} = \begin{cases} 1 & \text{if } |A \cap \{x, y\}| = 1 \\ 0 & \text{otherwise .} \end{cases}$$

Take $N$ to be even and $R \geq 2$ and define the submodular function $F^{\text{lb}} = F_1^{\text{lb}} + \cdots + F_R^{\text{lb}}$, where

$$F_1^{\text{lb}} = G_{12} + G_{34} + \cdots + G_{(N-1)N} \qquad F_2^{\text{lb}} = G_{23} + G_{45} + \cdots + G_{N1}$$

and $F_r^{\text{lb}} = 0$ for all $r \geq 3$. The optimal solution to the best-approximation problem is the all zeros vector.

**Lemma 13.** *The cosine of the Friedrichs angle between $\mathcal{A}$ and* $\text{aff}(\mathcal{B}^{lb})$ *is*

$$c_F(\mathcal{A}, \text{aff}(\mathcal{B}^{lb}))^2 = 1 - \tfrac{1}{R}\left(1 - \cos\left(\tfrac{2\pi}{N}\right)\right).$$

Around the optimal solution 0, the polyhedra $\mathcal{A}$ and $\mathcal{B}^{\text{lb}}$ behave like subspaces, and it is possible to pick initializations $a_0 \in \mathcal{A}$ and $b_0 \in \mathcal{B}^{\text{lb}}$ such that the Friedrichs angle exactly determines the rate of convergence. That means $1 - 1/\kappa_*^2 = c_F(\mathcal{A}, \text{aff}(\mathcal{B}^{\text{lb}}))^2$, and

$$\|a_k\| = (1 - \tfrac{1}{R}(1 - \cos(\tfrac{2\pi}{N})))^k\|a_0\| \quad and \quad \|b_k\| = (1 - \tfrac{1}{R}(1 - \cos(\tfrac{2\pi}{N})))^k\|b_0\|.$$

Bounding $1 - \cos(x) \leq \frac{1}{2}x^2$ leads to the following lower bound on the rate.

**Theorem 14.** *There exists a decomposed function $F^{lb}$ and initializations for which the convergence rate of AP is at least* $1 - \frac{2\pi^2}{N^2R}$.

This theoretical bound can also be observed empirically (Figure 3 in Appendix B).

## 5 Convergence of the Primal Objective

We have shown that AP generates a sequence of points $\{a_k\}_{k \geq 0}$ and $\{b_k\}_{k \geq 0}$ in $\mathbb{R}^{NR}$ such that $(a_k, b_k) \to (a_*, b_*)$ linearly, where $(a_*, b_*)$ minimizes the objective in Problem (P4). In this section, we show that this result also implies the linear convergence of the objective in Problem (P3) and of the original discrete objective in Problem (P1). The proofs may be found in Appendix C.

Define the matrix $\Gamma = -R^{1/2}S$, where $S$ is the matrix defined in Equation (5). Multiplication by $\Gamma$ maps a vector $(w_1, \ldots, w_R)$ to $-\sum_r w_r$, where $w_r \in \mathbb{R}^N$ for each $r$. Set $x_k = \Gamma b_k$ and $x_* = \Gamma b_*$. As shown in Jegelka et al. [25], Problem (P3) is minimized by $x_*$.

**Proposition 15.** *We have $f(x_k) + \frac{1}{2}\|x_k\|^2 \to f(x_*) + \frac{1}{2}\|x_*\|^2$ linearly with rate* $1 - \frac{1}{N^2R^2}$.

This linear rate of convergence translates into a linear rate for the original discrete problem.

**Theorem 16.** *Choose $A_* \in \arg\min_{A \subseteq V} F(A)$. Let $A_k$ be the suplevel set of $x_k$ with smallest value of $F$. Then $F(A_k) \to F(A_*)$ linearly with rate* $1 - \frac{1}{2N^2R^2}$.

# 6 Discussion

In this work, we analyze projection methods for parallel SFM and give upper and lower bounds on the linear rate of convergence. This means that the number of iterations required for an accuracy of $\epsilon$ is logarithmic in $1/\epsilon$, not linear as in previous work [35]. Our rate is uniform over all submodular functions. Moreover, our proof highlights how the number $R$ of components and the facial structure of $\mathcal{B}$ affect the convergence rate. These insights may serve as guidelines when working with projection algorithms and aid in the analysis of special cases. For example, reducing $R$ is often possible. Any collection of $F_r$ that have disjoint support, such as the cut functions corresponding to the rows or columns of a grid graph, can be grouped together without making the projection harder.

Our analysis also shows the effects of additional properties of $F$. For example, suppose that $F$ is *separable*, that is, $F(V) = F(S) + F(V \backslash S)$ for some nonempty $S \subsetneq V$. Then the subsets $A_{rm} \subseteq V$ defining the relevant faces of $\mathcal{B}$ satisfy either $A_{rm} \subseteq S$ or $A_{rm} \subseteq S^c$ [2]. This makes $G$ in Section 3.4 disconnected, and as a result, the $N$ in Theorem 12 gets replaced by $\max\{|S|, |S^c|\}$ for an improved rate. This applies without the user needing to know $S$ when running the algorithm.

A number of future directions suggest themselves. For example, Jegelka et al. [25] also considered the related Douglas–Rachford (DR) algorithm. DR between subspaces converges linearly with rate $c_F$ [6], as opposed to $c_F^2$ for AP. We suspect that our approach may be modified to analyze DR between polyhedra. Further questions include the extension to cyclic updates (instead of parallel ones), multiple polyhedra, and stochastic algorithms.

**Acknowledgments.** We would like to thank Mădălina Persu for suggesting the use of Cheeger's inequality. This research is supported in part by NSF CISE Expeditions Award CCF-1139158, LBNL Award 7076018, and DARPA XData Award FA8750-12-2-0331, and gifts from Amazon Web Services, Google, SAP, The Thomas and Stacey Siebel Foundation, Apple, C3Energy, Cisco, Cloudera, EMC, Ericsson, Facebook, GameOnTalis, Guavus, HP, Huawei, Intel, Microsoft, NetApp, Pivotal, Splunk, Virdata, VMware, WANdisco, and Yahoo!. This work is supported in part by the Office of Naval Research under grant number N00014-11-1-0688, the US ARL and the US ARO under grant number W911NF-11-1-0391, and the NSF under grant number DGE-1106400.

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
