[Supplementary Material]

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

# A  Upper Bound Results

## A.1  Proof of Theorem 2

For the proof of this theorem, we will need the fact that projection maps are firmly nonexpansive, that is, for a closed convex nonempty subset $C \subseteq \mathbb{R}^D$, we have
$$\|\Pi_C x - \Pi_C y\|^2 + \|(x - \Pi_C x) - (y - \Pi_C y)\|^2 \leq \|x - y\|^2$$
for all $x, y \in \mathbb{R}^D$. Now, suppose that $\kappa_* < \infty$. Let $e = \Pi_E p_k$ and note that $v = \Pi_Q e - e$ and that $\Pi_Q e \in H$. We have

$$
\begin{aligned}
\kappa_*^{-2} d(p_k, E)^2 &\leq d(p_k, Q')^2 \\
&\leq \|p_k - \Pi_Q p_k + v\|^2 \\
&\leq \|(p_k - \Pi_Q p_k) - (e - \Pi_Q e)\|^2 \\
&\leq \|p_k - e\|^2 - \|\Pi_Q p_k - \Pi_Q e\|^2 \\
&\leq d(p_k, E)^2 - d(q_k, H)^2.
\end{aligned}
$$

It follows that $d(q_k, H) \leq (1 - \kappa_*^{-2})^{1/2} d(p_k, E)$. Similarly, we have $d(p_{k+1}, E) \leq (1 - \kappa_*^{-2})^{1/2} d(q_k, H)$. When combining these, induction shows that

$$d(p_k, E) \leq (1 - \kappa_*^{-2})^k d(p_0, E)$$
$$d(q_k, H) \leq (1 - \kappa_*^{-2})^k d(q_0, H).$$

As shown in [3, Theorem 3.3], the above implies that $p_k \to p_* \in E$ and $q_k \to q_* \in H$ and that

$$\|p_k - p_*\| \leq 2\|p_0 - p_*\|(1 - \kappa_*^{-2})^k$$
$$\|q_k - q_*\| \leq 2\|q_0 - q_*\|(1 - \kappa_*^{-2})^k.$$

## A.2  Connection Between $\kappa$ and $c_F$ in the Subspace Case

In this section, we introduce a simple lemma connecting $\kappa$ and $c_F$ in the case of subspaces $U$ and $V$. We will use this lemma in several subsequent proofs.

**Lemma 17.** *Let $U$ and $V$ be subspaces and suppose $u \in U \cap (U \cap V)^{\perp}$ and that $u \neq 0$. Then*

*(a)* $\|\Pi_V u\| \leq c_F(U, V)\|u\|$
*(b)* $\kappa(u) \leq (1 - c_F(U, V)^2)^{-1/2}$
*(c)* $\kappa(u) = (1 - c_F(U, V)^2)^{-1/2}$ *if and only if* $\|\Pi_V u\| = c_F(U, V)\|u\|$.

*Proof.* Part (a) follows from the definition of $c_F$. Indeed,

$$c_F(U, V) \geq \frac{u^{\top}(\Pi_V u)}{\|u\|\|\Pi_V u\|} = \frac{\|\Pi_V u\|^2}{\|u\|\|\Pi_V u\|} = \frac{\|\Pi_V u\|}{\|u\|}.$$

Part (b) follows from Part (a) and the observation that $\kappa(u) = (1 - \|\Pi_V u\|^2/\|u\|^2)^{-1/2}$. Part (c) follows from the same observation. $\qquad\square$

## A.3  Proof of Lemma 3

It suffices to prove the statement for $p \in P \backslash E$. For $p \in P \backslash E$, define $q = \Pi_{Q'} p$, $e = \Pi_E q$, and $p'' = \Pi_{[p,e]} q$, where $[p, e]$ denotes the line segment between $p$ and $e$ (which is contained in $P$ by convexity). See Figure 2 for a graphical depiction. If $q \in E$, then $\kappa(p) = 1$. So we may assume that $q \notin E$ which also implies that $d(p'', E) > 0$ and $d(\Pi_P q, E) > 0$. We have

$$\kappa(p) = \frac{d(p, E)}{d(p, Q')} \leq \frac{\|p - e\|}{\|p - q\|} \leq \frac{\|q - e\|}{\|q - p''\|} \leq \frac{d(q, E)}{d(q, P)} = \kappa(q). \tag{8}$$

The first inequality holds because $d(p, E) \leq \|p - e\|$ and $d(p, Q') = \|p - q\|$. The middle inequality holds because the area of the triangle with vertices $p$, $q$, and $e$ can be expressed as both $\frac{1}{2}\|p - e\|\|q - p''\|$ and $\frac{1}{2}\|p - q\|\|q - e\|\sin\theta$, where $\theta$ is the angle between vectors $p - q$ and $e - q$, so

$$\|p - e\|\|q - p''\| = \|p - q\|\|q - e\|\sin\theta \leq \|p - q\|\|q - e\|.$$

The third inequality holds because $\|q - e\| = d(q, E)$ and $\|q - p''\| \geq d(q, P)$. The chain of inequalities in Equation (8) prove the lemma.

Figure 2: Illustration of the proof of Lemma 3.

## A.4 Proof of Proposition 4

Suppose that $p \in P \backslash E$ (the case $q \in Q' \backslash E$ is the same), and let $e = \Pi_E p$. If $\kappa(p) = 1$, the statement is evident, so we may assume that $\kappa(p) > 1$. We will construct sequences of polyhedra

$$
\begin{array}{cccccc}
P & \supseteq & P_1 & \supseteq & \cdots & \supseteq & P_J \\
Q' & \supseteq & Q'_1 & \supseteq & \cdots & \supseteq & Q'_J
\end{array} .
$$

where $P_{j+1}$ is a face of $P_j$ and $Q'_{j+1}$ is a face of $Q'_j$ for $1 \leq j \leq J - 1$. Either $\dim(\mathrm{aff}(P_{j+1})) < \dim(\mathrm{aff}(P_j))$ or $\dim(\mathrm{aff}(Q'_{j+1})) < \dim(\mathrm{aff}(Q'_j))$ will hold. We will further define $E_j = P_j \cap Q'_j$, which will contain $e$, so that we can define

$$
\kappa_j(x) = \frac{d(x, E_j)}{\max\{d(x, P_j), d(x, Q'_j)\}}
$$

for $x \in \mathbb{R}^D \backslash E_j$ (this is just the function $\kappa$ defined for the polyhedra $P_j$ and $Q'_j$). Our construction will yield points $p_j \in P_j$, and $q_j \in Q'_j$ such that $p_j \in \mathrm{relint}(P_j) \backslash E_j$, $q_j \in \mathrm{relint}(Q'_j) \backslash E_j$, and $q_j = \Pi_{Q'_j} p_j$ for each $j$. Furthermore, we will have

$$
\kappa(p) \leq \kappa_1(p_1) \leq \cdots \leq \kappa_J(p_J). \tag{9}
$$

Now we describe the construction. For any $t \in [0, 1]$, define $p^t = (1 - t)p + te$ to be the point obtained by moving $p$ by the appropriate amount toward $e$. Note that $t \mapsto \kappa(p^t)$ is a nondecreasing function on the interval $[0, 1)$. Choose $\epsilon > 0$ sufficiently small so that every face of either $P$ or $Q'$ that intersects $B_\epsilon(e)$, the ball of radius $\epsilon$ centered on $e$, necessarily contains $e$. Now choose $0 \leq t_0 < 1$ sufficiently close to 1 so that $\|p^{t_0} - e\| < \epsilon$. It follows that $e$ is contained in the face of $P$ whose relative interior contains $p^{t_0}$. It further follows that $e$ is contained in the face of $Q'$ whose relative interior contains $\Pi_{Q'} p^{t_0}$ because

$$
\|\Pi_{Q'} p^{t_0} - e\| = \|\Pi_{Q'} p^{t_0} - \Pi_{Q'} e\| \leq \|p^{t_0} - e\| < \epsilon.
$$

To initialize the construction, set

$$
p_1 = p^{t_0}
$$
$$
q_1 = \Pi_{Q'} p^{t_0},
$$

and let $P_1$ and $Q'_1$ be the unique faces of $P$ and $Q'$ respectively such that $p_1 \in \mathrm{relint}(P_1)$ and $q_1 \in \mathrm{relint}(Q'_1)$ (the relative interiors of the faces of a polyhedron partition that polyhedron [8, Theorem 2.2]). Note that $q_1 \notin E$ because $\kappa(p_1) \geq \kappa(p) > 1$. Note that $e \in E_1 = P_1 \cap Q'_1$ so that

$$
\kappa(p) \leq \kappa(p_1) = \frac{d(p_1, E)}{d(p_1, Q')} = \frac{\|p_1 - e\|}{\|p_1 - q_1\|} = \frac{d(p_1, E_1)}{d(p_1, Q'_1)} = \kappa_1(p_1).
$$

Now, inductively assume that we have defined $P_j$, $Q'_j$, $p_j$, and $q_j$ satisfying the stated properties. Generate the sequences $\{x_k\}_{k\geq 0}$ and $\{y_k\}_{k\geq 0}$ with $x_k \in P_j$ and $y_k \in Q'_j$ by running AP between the polyhedra $P_j$ and $Q'_j$ initialized with $x_0 = p_j$. There are two possibilities, either $x_k \in \mathrm{relint}(P_j)$ and $y_k \in \mathrm{relint}(Q'_j)$ for every $k$, or there is some $k$ for which either $x_k \notin \mathrm{relint}(P_j)$ or $y_k \notin \mathrm{relint}(Q'_j)$. Note that $P_j$ and $Q'_j$ intersect and that AP between them will not terminate after a finite number of steps.

Suppose that $x_k \in \mathrm{relint}(P_j)$ and $y_k \in \mathrm{relint}(Q'_j)$ for every $k$. Then set $J = j$ and terminate the procedure. Otherwise, choose $k'$ such that either $x_{k'} \notin \mathrm{relint}(P_j)$ or $y_{k'} \notin \mathrm{relint}(Q'_j)$. Now set $p_{j+1} = x_{k'}$ and $q_{j+1} = y_{k'}$. Let $P_{j+1}$ and $Q'_{j+1}$ be the unique faces of $P_j$ and $Q'_j$ respectively such that $p_{j+1} \in \mathrm{relint}(P_{j+1})$ and $q_{j+1} \in \mathrm{relint}(Q'_{j+1})$. Note that $p_{j+1}, q_{j+1} \notin E_{j+1} = P_{j+1} \cap Q'_{j+1}$ and $e \in E_{j+1}$. We have

$$\kappa_j(p_j) < \kappa_j(p_{j+1}) = \frac{d(p_{j+1}, E_j)}{d(p_{j+1}, Q'_j)} = \frac{d(p_{j+1}, E_j)}{\|p_{j+1} - q_{j+1}\|} \leq \frac{d(p_{j+1}, E_{j+1})}{d(p_{j+1}, Q'_{j+1})} = \kappa_{j+1}(p_{j+1}).$$

The preceding work shows the inductive step. Note that if $P_{j+1} \neq P_j$ then $\dim(\mathrm{aff}(P_{j+1})) < \dim(\mathrm{aff}(P_j))$ and if $Q'_{j+1} \neq Q'_j$ then $\dim(\mathrm{aff}(Q'_{j+1})) < \dim(\mathrm{aff}(Q'_j))$. One of these will hold, so the induction will terminate after a finite number of steps.

We have produced the sequence in Equation (9) and we have created $p_J$, $P_J$, and $Q'_J$ such that AP between $P_J$ and $Q'_J$, when initialized at $p_J$, generates the same sequence of points as AP between $\mathrm{aff}(P_J)$ and $\mathrm{aff}(Q'_J)$. Using this fact, along with [12, Theorem 9.3], we see that $\Pi_{\mathrm{aff}(P_J)\cap\mathrm{aff}(Q'_J)}p_J \in E_J$. Using this, along with Lemma 17(b), we see that

$$\kappa_J(p_J) \leq (1 - c_F(\mathrm{aff}_0(P_J), \mathrm{aff}_0(Q'_J))^2)^{-1/2}. \tag{10}$$

Equations (10) and (9) prove the result. Note that $P_J$ and $Q'_J$ are faces of $P$ and $Q'$ respectively. We can switch between faces of $Q'$ and faces of $Q$ because doing so amounts to translating by $v$ which does not affect the angles.

## A.5 Proof of Lemma 6

We have

$$c_F(\mathrm{null}(S), \mathrm{null}(T)) = c_F(\mathrm{range}(S^\top)^\perp, \mathrm{range}(T^\top)^\perp)$$
$$= c_F(\mathrm{range}(S^\top), \mathrm{range}(T^\top)),$$

where the first equality uses the fact that $\mathrm{null}(W) = \mathrm{range}(W^\top)^\perp$ for matrices $W$, and the second equality uses the fact that $c_F(U^\perp, V^\perp) = c_F(U, V)$ for subspaces $U$ and $V$ [6, Fact 2.3].

Let $S^\top$ and $T^\top$ have dimensions $D \times J$ and $D \times K$ respectively, and let $X$ and $Y$ be the subspaces spanned by the columns of $S^\top$ and $T^\top$ respectively. Without loss of generality, assume that $J \leq K$. Let $\sigma_1 \geq \cdots \geq \sigma_J$ be the singular values of $ST^\top$ with corresponding left singular vectors $u_1, \ldots, u_J$ and right singular vectors $v_1, \ldots, v_J$. Let $x_j = S^\top u_j$ and let $y_j = T^\top v_j$ for $1 \leq j \leq J$. By definition, we can write

$$\sigma_j = \max_{u,v}\{u^\top ST^\top v \mid u \perp \mathrm{span}(u_1, \ldots, u_{j-1}), v \perp \mathrm{span}(v_1, \ldots, v_{j-1}), \|u\| = 1, \|v\| = 1\}.$$

Since the $\{u_j\}_j$ are orthonormal, so are the $\{x_j\}_j$. Similarly, since the $\{v_j\}_j$ are orthonormal, so are the $\{y_j\}_j$. Suppose that all of the singular values of $ST^\top$ equal one. Then we must have $x_j = y_j$ for each $j$, which implies that $X \subseteq Y$, and so $c_F(X, Y) = 0$.

Now suppose that $\sigma_1 = \cdots = \sigma_\ell = 1$, and $\sigma_{\ell+1} \neq 1$. It follows that

$$X \cap Y = \mathrm{span}(x_1, \ldots, x_\ell) = \mathrm{span}(y_1, \ldots, y_\ell),$$

and so

$$\sigma_{\ell+1} = \sup_{u,v}\{u^\top ST^\top v \mid u \in \mathrm{span}(u_1, \ldots, u_\ell)^\perp, v \in \mathrm{span}(v_1, \ldots, v_\ell)^\perp, \|u\| = 1, \|v\| = 1\}$$
$$= \sup_{x,y}\{x^\top y \mid x \in X \cap (X \cap Y)^\perp, y \in Y \cap (X \cap Y)^\perp, \|x\| = 1, \|y\| = 1\}$$
$$= c_F(X, Y).$$

## A.6 Cheeger's Inequality

For an overview of spectral graph theory, see Chung [9]. We state Cheeger's inequality below.

Let $G$ be a weighted, connected graph with vertex set $V_G$ and edge weights $(w_{ij})_{i,j \in V_G}$. Define the weighted degree of a vertex $i$ to be $\delta_i = \sum_{j \neq i} w_{ij}$, define the volume of a subset of vertices to be the sum of their weighted degrees, $\text{vol}(U) = \sum_{i \in U} \delta_i$, and define the size of the cut between $U$ and its complement $U^c$ to be the sum of the weights of the edges between $U$ and $U^c$,

$$|E(U, U^c)| = \sum_{i \in U, j \in U^c} w_{ij}.$$

The Cheeger constant is defined as

$$h_G = \min_{\emptyset \neq U \subsetneq V_G} \frac{|E(U, U^c)|}{\min(\text{vol}(U), \text{vol}(U^c))}.$$

Let $L$ be the unnormalized Laplacian of $G$, i.e. the $|V_G| \times |V_G|$ matrix whose entries are defined by

$$L_{ij} = \begin{cases} -w_{ij} & i \neq j \\ \delta_i & \text{otherwise} \end{cases}.$$

Let $D$ be the $|V_G| \times |V_G|$ diagonal matrix defined by $D_{ii} = \delta_i$. Then $\mathcal{L} = D^{-1/2} L D^{-1/2}$ is the normalized Laplacian. Let $\lambda_2(\mathcal{L})$ denote the second smallest eigenvalue of $\mathcal{L}$ (since $G$ is connected, there will be exactly one eigenvalue equal to zero).

**Theorem 18** (Cheeger's inequality). *We have $\lambda_2(\mathcal{L}) \geq \frac{h_G^2}{2}$.*

## A.7 Proof of Lemma 10

*Proof.* We have

$$\min(\text{vol}(U), \text{vol}(U^c)) \leq \frac{1}{2} \text{vol}(V_G)$$

$$= \frac{1}{2} \sum_{(r,m)} \left( \sum_{(r',m') \neq (r,m)} |A_{rm} \cap A_{r'm'}| \right)$$

$$= \frac{1}{2} \sum_{(r,m)} (R-1)|A_{rm}|$$

$$= \frac{1}{2} NR(R-1).$$

Since $G$ is connected, for any nonempty set $U \subsetneq V_G$, there must be some element $v \in V$ (here $V$ is the ground set of our submodular function $F$, not the set of vertices $V_G$) such that $v \in A_{r_1 m_1} \cap A_{r_2 m_2}$ for some $(r_1, m_1) \in U$ and $(r_2, m_2) \in U^c$. Suppose that $v$ appears in $k$ of the subsets of $V$ indexed by elements of $U$ and in $R-k$ of the subsets of $V$ indexed by elements of $U^c$. Then

$$|E(U, U^c)| \geq k(R-k) \geq R - 1.$$

It follows that

$$h_G \geq \frac{R-1}{\frac{1}{2}NR(R-1)} = \frac{2}{NR}.$$

$\square$

It follows from Theorem 18 that $\lambda_2(\mathcal{L}) \geq \frac{2}{N^2 R^2}$.

# B  Results for the Lower Bound

## B.1  Some Helpful Results

In Lemma 19, we show how AP between subspaces $U$ and $V$ can be initialized to exactly achieve the worst-case rate of convergence. Then in Corollary 20, we show that if subsets $U'$ and $V'$ look like subspaces $U$ and $V$ near the origin, we can initialize AP between $U'$ and $V'$ to achieve the same worst-case rate of convergence.

**Lemma 19.** *Let $U$ and $V$ be subspaces with $U \not\subseteq V$ and $V \not\subseteq U$. Then there exists some nonzero point $u_0 \in U \cap (U \cap V)^\perp$ such that when we initialize AP at $u_0$, the resulting sequences $\{u_k\}_{k \geq 0}$ and $\{v_k\}_{k \geq 0}$ satisfy*

$$\|u_k\| = c_F(U,V)^{2k}\|u_0\|$$
$$\|v_k\| = c_F(U,V)^{2k}\|v_0\|.$$

*Proof.* Find $u_* \in U \cap (U \cap V)^\perp$ and $v_* \in V \cap (U \cap V)^\perp$ with $\|u_*\| = 1$ and $\|v_*\| = 1$ such that $u_*^\top v_* = c_F(U,V)$, which we can do by compactness. By Lemma 17(a),

$$c_F(U,V) = v_*^\top u_* = v_*^\top \Pi_V u_* \leq \|\Pi_V u_*\| \leq c_F(U,V).$$

Set $u_0 = u_*$ and generate the sequences $\{u_k\}_{k \geq 0}$ and $\{v_k\}_{k \geq 0}$ via AP. Since $\|\Pi_V u_0\| = c_F(U,V)$, Lemma 17(c) implies that $\kappa(u_0) = (1 - c_F(U,V)^2)^{-1/2}$. Since $\kappa$ attains its maximum at $u_0$, Lemma 3 implies that $\kappa$ attains the same value at every element of the sequences $\{u_k\}_{k \geq 0}$ and $\{v_k\}_{k \geq 0}$. Therefore, Lemma 17(c) implies that $\|\Pi_V u_k\| = c_F(U,V)\|u_k\|$ and $\|\Pi_U v_k\| = c_F(U,V)\|v_k\|$ for all $k$. This proves the lemma. $\square$

**Corollary 20.** *Let $U$ and $V$ be subspaces with $U \not\subseteq V$ and $V \not\subseteq U$. Let $U' \subseteq U$ and $V' \subseteq V$ be subsets such that $U' \cap B_\epsilon(0) = U \cap B_\epsilon(0)$ and $V' \cap B_\epsilon(0) = V \cap B_\epsilon(0)$ for some $\epsilon > 0$. Then there is a point $u_0' \in U'$ such that the sequences $\{u_k'\}_{k \geq 0}$ and $\{v_k'\}_{k \geq 0}$ generated by AP between $U'$ and $V'$ initialized at $u_0'$ satisfy*

$$\|u_k'\| = c_F(U,V)^{2k}\|u_0'\|$$
$$\|v_k'\| = c_F(U,V)^{2k}\|v_0'\|.$$

*Proof.* Use Lemma 19 to choose some nonzero $u_0 \in U \cap (U \cap V)^\perp$ satisfying this property. Then set $u_0' = \frac{\epsilon}{\|u_0\|}u_0$. $\square$

## B.2 Proof of Lemma 13

Observe that we can write

$$\text{aff}(\mathcal{B}^{\text{lb}}) = \{(s_1, -s_1, \ldots, s_{\frac{N}{2}}, -s_{\frac{N}{2}}, -t_{\frac{N}{2}}, t_1, -t_1, \ldots, t_{\frac{N}{2}}, 0, \ldots, 0, \ldots, 0, \ldots, 0) \mid s_i, t_j \in \mathbb{R}\}.$$

We can write $\text{aff}(\mathcal{B}^{\text{lb}})$ as the nullspace of the matrix

$$T_{\text{lb}} = \begin{pmatrix} T_{\text{lb},1} & & & & \\ & T_{\text{lb},2} & & & \\ & & I_N & & \\ & & & \ddots & \\ & & & & I_N \end{pmatrix},$$

where the $N \times N$ identity matrix $I_N$ is repeated $R-2$ times and where $T_{\text{lb},1}$ and $T_{\text{lb},2}$ are the $\frac{N}{2} \times N$ matrices

$$T_{\text{lb},1} = \frac{1}{\sqrt{2}}\begin{pmatrix} 1 & 1 & & & \\ & & 1 & 1 & \\ & & & & \ddots & \\ & & & & & 1 & 1 \end{pmatrix} \qquad T_{\text{lb},2} = \frac{1}{\sqrt{2}}\begin{pmatrix} 1 & 1 & & & \\ & & 1 & 1 & \\ & & & & \ddots & \\ 1 & & & & & 1 \end{pmatrix}.$$

Recall that we can write $\mathcal{A}$ as the nullspace of the matrix $S$ defined in Equation (5). It follows from Lemma 6 that $c_F(\mathcal{A}, \text{aff}(\mathcal{B}^{\text{lb}}))$ equals the largest singular value of $ST_{\text{lb}}^\top$ that is less than one. We have

$$ST_{\text{lb}}^\top = \tfrac{1}{\sqrt{R}}\begin{pmatrix} T_{\text{lb},1}^\top & T_{\text{lb},2}^\top & I_N & \cdots & I_N \end{pmatrix}.$$

We can permute the columns of $ST_{\text{lb}}^\top$ without changing the singular values, so $c_F(\mathcal{A}, \text{aff}(\mathcal{B}^{\text{lb}}))$ equals the largest singular value of

$$\tfrac{1}{\sqrt{R}}\begin{pmatrix} T_{\text{lb},0}^\top & I_N & \cdots & I_N \end{pmatrix},$$

Figure 3: We run five trials of AP between $\mathcal{A}$ and $\mathcal{B}^{\text{lb}}$ with random initializations, where $N = 10$ and $R = 10$. For each trial, we plot the ratios $d(a_{k+1}, E)/d(a_k, E)$, where $E = \mathcal{A} \cap \mathcal{B}^{\text{lb}}$ is the optimal set. The red line shows the theoretical lower bound of $1 - \frac{1}{R}(1 - \cos(\frac{2\pi}{N}))$ on the worst-case rate of convergence.

that is less than one, where $T_{\text{lb},0}$ is the $N \times N$ circulant matrix

$$
T_{\text{lb},0} = \frac{1}{\sqrt{2}} \begin{pmatrix} 1 & 1 & & & \\ & 1 & 1 & & \\ & & & \ddots & \\ & & & 1 & 1 \\ 1 & & & & 1 \end{pmatrix}.
$$

Therefore, $c_F(\mathcal{A}, \text{aff}(\mathcal{B}^{\text{lb}}))^2$ equals the largest eigenvalue of

$$
\frac{1}{R} \begin{pmatrix} T_{\text{lb},0}^\top & I_N & \cdots & I_N \end{pmatrix} \begin{pmatrix} T_{\text{lb},0}^\top & I_N & \cdots & I_N \end{pmatrix}^\top = \frac{1}{R} \left( T_{\text{lb},0}^\top T_{\text{lb},0} + (R-2)I_N \right)
$$

that is less than one. Therefore, it suffices to examine the $N \times N$ circulant matrix

$$
T_{\text{lb},0}^\top T_{\text{lb},0} = \frac{1}{2} \begin{pmatrix} 2 & 1 & & & 1 \\ 1 & 2 & & & \\ & & \ddots & & \\ & & & 2 & 1 \\ 1 & & & 1 & 2 \end{pmatrix}.
$$

The eigenvalues of $T_{\text{lb},0}^\top T_{\text{lb},0}$ are given by $\lambda_j = 1 + \cos\left(\frac{2\pi j}{N}\right)$ for $0 \le j \le N - 1$ (see Gray [18, Section 3.1] for a derivation). Therefore,

$$
c_F(\mathcal{A}, \text{aff}(\mathcal{B}^{\text{lb}}))^2 = 1 - \frac{1}{R}(1 - \cos(\frac{2\pi}{N})).
$$

## B.3  Lower Bound Illustration

The proof of Theorem 14 shows that there is some $a_0 \in \mathcal{A}$ such that when we initialize AP between $\mathcal{A}$ and $\mathcal{B}^{\text{lb}}$ at $a_0$, we generate a sequence $\{a_k\}_{k \ge 0}$ satisfying

$$
d(a_k, E) = (1 - \frac{1}{R}(1 - \cos(\frac{2\pi}{N})))^k d(a_0, E),
$$

where $E = \mathcal{A} \cap \mathcal{B}^{\text{lb}}$ is the optimal set. In Figure 3, we plot the theoretical bound in red, and in blue the successive ratios $d(a_{k+1}, E)/d(a_k, E)$ for five runs of AP between $\mathcal{A}$ and $\mathcal{B}^{\text{lb}}$ with random initializations. Had we initialized AP at $a_0$, the successive ratios would exactly equal $1 - \frac{1}{R}(1 - \cos(\frac{2\pi}{N}))$. The plot of these ratios would coincide with the red line in Figure 3.

Figure 3 illustrates that the empirical behavior of AP between $\mathcal{A}$ and $\mathcal{B}^{\text{lb}}$ is often similar to the worst-case behavior, even when the initialization is random. When we initialize AP randomly, the successive ratios appear to increase to the lower bound and then remain constant. Figure 3 shows the case $N = 10$ and $R = 10$, but the plot looks similar for other $N$ and $R$.

We also note that the graph corresponding to our lower bound example actually achieves a Cheeger constant similar to the one used in Lemma 10.

## C   Results for Convergence of the Primal and Discrete Problems

### C.1   Proof of Proposition 15

First, suppose that $s \in B(F)$. Let $A = \{n \in V \,|\, s_n \geq 0\}$ be the set of indices on which $s$ is nonnegative. Then we have

$$\|s\| \leq \|s\|_1 = 2s(A) - s(V) \leq 3F_{\max}. \tag{11}$$

Recall that we defined $F_{\max} = \max_A |F(A)|$. Now, we show that $f(x_k) + \frac{1}{2}\|x_k\|^2$ converges to $f(x_*) + \frac{1}{2}\|x_*\|^2$ linearly with rate $1 - \frac{1}{N^2 R^2}$. We will use Equation (11) to bound the norms of $x_k$ and $x_*$, both of which lie in $-B(F)$. We will also use the fact that $\|x_k - x_*\| \leq \|\Gamma\|\|b_k - b_*\| \leq \sqrt{R}\|b_k - b_*\|$. Finally, we will use the proof of Theorem 12 to bound $\|b_k - b_*\|$. First, we bound the difference between the squared norms using convexity. We have

$$\begin{aligned}
\tfrac{1}{2}\|x_k\|^2 - \tfrac{1}{2}\|x_*\|^2 &\leq x_k^\top (x_k - x_*) \\
&\leq \|x_k\|\|x_k - x_*\| \\
&\leq 3F_{\max}\sqrt{R}\|b_k - b_*\| \\
&\leq 6F_{\max}\sqrt{R}\|b_0 - b_*\|(1 - \tfrac{1}{N^2 R^2})^k.
\end{aligned} \tag{12}$$

Next, we bound the difference in Lovász extensions. Choose $s \in \arg\max_{s \in B(F)} s^\top x_k$. Then

$$\begin{aligned}
f(x_k) - f(x_*) &\leq s^\top (x_k - x_*) \\
&\leq \|s\|\|x_k - x_*\| \\
&\leq 3F_{\max}\sqrt{R}\|b_k - b_*\| \\
&\leq 6F_{\max}\sqrt{R}\|b_0 - b_*\|(1 - \tfrac{1}{N^2 R^2})^k.
\end{aligned} \tag{13}$$

Combining the bounds (12) and (13), we find that

$$(f(x_k) + \tfrac{1}{2}\|x_k\|^2) - (f(x_*) + \tfrac{1}{2}\|x_*\|^2) \leq 12F_{\max}\sqrt{R}\|b_0 - b_*\|(1 - \tfrac{1}{N^2 R^2})^k. \tag{14}$$

### C.2   Proof of Theorem 16

We will make use of the following result, shown in [2, Proposition 10.5] and stated below for convenience.

**Proposition 21.** *Let $(w, s) \in \mathbb{R}^N \times B(F)$ be a pair of primal-dual candidates for the minimization of $\frac{1}{2}\|w\|^2 + f(w)$, with duality gap $\epsilon = \frac{1}{2}\|w\|^2 + f(w) + \frac{1}{2}\|s\|^2$. Then if $A$ is the suplevel set of $w$ with smallest value of $F$, then*

$$F(A) - s_-(V) \leq \sqrt{N\epsilon/2}.$$

Using this result in our setting, recall that by definition $A_k$ is the set of the form $\{n \in V \,|\, (x_k)_n \geq c\}$ for some constant $c$ with smallest value of $F(\{n \in V \,|\, (x_k)_n \geq c\})$.

Let $(w_*, s_*) \in \mathbb{R}^N \times B(F)$ be a primal-dual optimal pair for the left-hand version of Problem (P3). The dual of this minimization problem is the projection problem $\min_{s \in B(F)} \frac{1}{2}\|s\|^2$. From [2, Propo-

sition 10.5], we see that

$$F(A_k) - F(A_*) \leq F(A_k) - (s_*)_-(V)$$
$$\leq \sqrt{\tfrac{N}{2}\left((f(x_k) + \tfrac{1}{2}\|x_k\|^2) - (f(x_*) + \tfrac{1}{2}\|x_*\|^2)\right)}$$
$$\leq \sqrt{6F_{\max}NR^{1/2}\|b_0 - b_*\|}\,(1 - \tfrac{1}{N^2R^2})^{k/2}$$
$$\leq \sqrt{6F_{\max}NR^{1/2}\|b_0 - b_*\|}\,(1 - \tfrac{1}{2N^2R^2})^{k},$$

where the third inequality uses the proof of Proposition 15. The second inequality relies on Bach [2, Proposition 10.5], which states that a duality gap of $\epsilon$ for the left-hand version of Problem (P3) turns into a duality gap of $\sqrt{N\epsilon/2}$ for the original discrete problem. If our algorithm converged with rate $\frac{1}{k}$, this would translate to a rate of $\frac{1}{\sqrt{k}}$ for the discrete problem. But fortunately, our algorithm converges linearly, and taking a square root preserves linear convergence.

## C.3 Running times

Theorem 16 implies that the number of iterations required for an accuracy of $\epsilon$ is at most

$$2N^2R^2 \log\left(\frac{\sqrt{6F_{\max}NR^{1/2}\|b_0 - b_*\|}}{\epsilon}\right). \tag{15}$$

Each iteration involves minimizing each of the $F_r$ separately. For comparison, the number of iterations required in Stobbe and Krause [35] is

$$24\sqrt{N}R\frac{F_{\max}}{\epsilon}.$$

The dependence of this algorithm on $N$ and $R$ is better, but its dependence on $F_{\max}/\epsilon$ is worse. For example, to obtain the exact discrete solution, we need $\epsilon < \min_{S,T}|F(S) - F(T)|$. This is one for integer-valued functions (in which case the lower rate may be desirable), but can otherwise become very small. The constant $F_{\max}$ can be of order $O(N)$ in general (or even larger if the function becomes very negative). For empirical comparisons, we refer the reader to [25].

The running times of the combinatorial algorithm by Kolmogorov [29] apply to *integer-valued* functions (as opposed to the generic ones above) and range from $O((N + R)^2 \log F_{\max})$ for cuts to $O((N + Q^2R)(N + Q^2R + QR\tau_2) \log F_{\max})$, where $Q \leq N$ is the maximal cardinality of the support of any $F_r$, and $\tau_2$ is the time required to minimize a simple function. This is better than (15) if $Q$ is a small constant, and worse as $Q$ gets closer to $N$.

For comparison, if not exploiting decomposition, one may use combinatorial algorithms, the Frank-Wolfe algorithm (conditional gradient descent), or a subgradient method. The combinatorial algorithm by Orlin [34] has a running time of $O(N^5\tau_1 + N^6)$, and the algorithm by Iwata [23] (for integer-valued functions) has a running time of $O((N^4\tau_1 + N^5) \log F_{\max})$, where $\tau_1$ is the time required to evaluate $F$. For an accuracy of $\epsilon$ in the discrete objective, Frank-Wolfe will take $64N\frac{F_{\max}}{\epsilon^2}$ iterations, each taking time $O(N \log N)$. The subgradient method behaves similarly.