[Reviews · NeurIPS 2014]

Submitted by Assigned_Reviewer_24

The authors present a proof of a linear convergence rate for an alternating projection scheme that seeks to minimize a decomposable submodular function. The paper is well written tough, due to the length of the proofs, the background material and motivation is a bit terse in spots. The paper makes several contributions to the theory of projection algorithms, and defers to previous work for experimental results and applications and also provides a concrete example of a submodular function that achieves the predicted worst case rate. This seems like a solid theoretical contribution and similar techniques may be useful for the analysis of other projection-based algorithms.
Summary: This is a purely theoretical paper that proves a linear convergence rate for the alternating projections algorithm for minimizing a sum of submodular functions.

Submitted by Assigned_Reviewer_35

Eq. (3): what if the maximization problem is unbounded?

Line 216: the phrase "intimately related" suggests that $\kappa^*$ completely determines the convergence rate. But what Theorem 2 shows is that $\kappa^*$ only upper bounds the convergence rate, which is a weaker property.

Line 324: "the analysis in this case subsumes the analysis in the general case". Can you elaborate this argument?

I am curious, would you get better bounds if you assumed that functions F_i depend only on variables in some small subset of V?

Suggestions on the presentation:
- you use the word "subspace" throughout the paper. This is a very general term, I would suggest to use "linear subspace" to remove potential ambiguities.

- in Corollary 8 you could add that this set can be equivalently described by equalities $s(A_i)=0$, then you wouldn't need matrix T'.

- In lines 305 and 310 you use aff(A_x). It could be better to use aff_0(A_x), since that's how the convergence results (e.g. Corollary 5) are formulated.

I think it is important to state in the abstract which specific method you analyze (with a reference), and your main result, i.e. the convergence rate that you obtain.
Summary: The submission analyzes the method of alternating projections for minimizing a sum of submodular functions proposed by Jegelka et al. [24]. It is shown that the method converges linearly with the rate at most 1-1/(N^2 R^2) where N is the number of variables and R is the number of submodular terms. The authors also give an example with the rate at least 1-O(1/N^2 R), which shows that the analysis is fairly tight.

I find this to be an interesting result with a rather non-trivial analysis. (The most interesting part, in my view, is how the authors bound the eigenvalues of the matrix in section 3.4.) I am thus happy to recommend this paper for acceptance.

Most proofs are given in the Appendices (in the supplementary material). I haven't checked them, but the claims look plausible to me.

Submitted by Assigned_Reviewer_42

This paper proves that the alternating projections method yield to linear convergence rate of the solution and the primal objective in optimizing decomposable submodular problems.

The paper is very technical but is well organized. The authors provide some intuitions and roadmap in the beginning of each section. The improvement from the original O(1/\epsilon) to O(log(1/\epsilon)) is substantial. These theoretical bounds provide good justifications for the good empirical results using AP on submodular problems as shown in [24].
Summary: Convincing theoretical results.
Author Feedback
Author rebuttal: We thank all reviewers for their positive feedback and will try to improve the paper according to the comments.
We would like to point out that this paper analyzes a new, alternative way of minimizing the class of decomposable submodular functions. Given the increasing number of applications of submodular functions in machine learning, this analysis may find wider applications.

responses to Reviewer 2:

1. Eq. (3): what if the maximization problem is unbounded?:
That’s a good point. We need to define the faces of the polyhedron P as the nonempty maximizers of linear functions over P.

2. Line 216 (kappa):
We will phrase this better.

3. Line 324: "the analysis in this case subsumes the analysis in the general case". Can you elaborate on this?:
Yes, we will elaborate here and we can elaborate in the paper as well. In the paper, we described a procedure for taking a collection of partitions {A_rm} of V (where r indexes a partition and m indexes the subsets of the partition) and constructing a weighted graph. When the resulting graph is connected, we showed that the square of the smallest nonzero eigenvalue of the graph Laplacian must be at least 2/(N^2 R^2).
When the resulting graph is not connected, say that the graph G has connected components G_1,...,G_J, then the smallest nonzero eigenvalue of the graph Laplacian of G equals the smallest nonzero eigenvalue of the graph Laplacian of G_j for some j. One can show that the same bound of 2/(N^2 R^2) holds for the square of this value:
Let V_j be the set of vertices of G_j. Now suppose that U \subset V_j, U \neq V_j, is a nonempty subset of vertices. Then, as before, the volume of U (within the graph G_j) satisfies
min{vol(U), vol(V_j\U)} <= NR(R−1)/2.
Since G_j is connected, there must be some element v in V such that v in A_{r1,m1} and v in A_{r2,m2} for some (r1,m1) in U and (r2,m2) in V_j\U. Note that the element v appears in the subsets corresponding to exactly R of the vertices of G (one for each partition), and these vertices are all connected to one another in G, so they must all appear in G_j. Suppose k of them appear in U and R−k of them appear in V_j\U. Then
|E(U,V_j\U)| >= k(R−k) >= R−1.
As in the paper, we can then show that the square of the smallest nonzero eigenvalue of the graph Laplacian of G_j is at least 2/(N^2 R^2). It follows then that the same lower bound applies to the square of the smallest nonzero eigenvalue of the graph Laplacian of G.

4. better bounds if functions F_r depend only on variables in some small subset of V?:
If there exists a partition of V into disjoint sets V_1,...,V_J such that the support of each F_r is contained in some V_j, then we can replace the N in our upper bound with max_j |V_j|. In general though, we do not immediately see a way to incorporate the assumption that each F_r has small support to obtain a better upper bound.

5. Suggestions on the presentation and abstract:
Thanks, those seem like good ideas. However, we think that including the matrix T' makes the presentation clearer.